# Effects of Landscape and Local Factors on the Diversity of Flower-Visitor Groups under an Urbanization Gradient, a Case Study in Wuhan, China

Mengyu He [1,2], Nan Ran [1], Huiqian Jiang [1], Zemin Han [1], Yuanyong Dian [1], Xiaoxia Li [3], Dong Xie [4,5], Peter A. Bowler [6] and Hui Wang [1,*]

1   College of Horticulture & Forestry Sciences, Hubei Engineering Technology Research Center for Forestry Information, Huazhong Agricultural University, Wuhan 430070, China; hmyforestry@webmail.hzau.edu.cn (M.H.); rannan@webmail.hzau.edu.cn (N.R.); jianghuiqian1023@163.com (H.J.); hzm@webmail.hzau.edu.cn (Z.H.); dianyuanyong@126.com (Y.D.)
2   Wuhan Dijiao Park Management Office, Garden and Forestry Bureau of Jiang'an District, Wuhan 430014, China
3   Institute of Wetland Research, Chinese Academy of Forestry, Beijing 100091, China; kindxiaoxia@163.com
4   Co-Innovation Center for Sustainable Forestry in Southern China, College of Biology and the Environment, Nanjing Forestry University, Nanjing 210037, China; xiedong0123@gmail.com
5   The National Wetland Ecosystem Field Station of Taihu Lake, National Forestry Administration, Suzhou 215107, China
6   Department of Ecology and Evolutionary Biology, University of California, Irvine, CA 92697-2525, USA; padowler@uci.edu
*   Correspondence: wanghui@mail.hzau.edu.cn; Tel.: +86-1877-113-2510

**Abstract:** Urbanization is one of the primary forces driving worldwide pollinator decline. Moderate urban expansion with appropriate green space planning can help in maintaining pollinator diversity and pollination service. We investigated the relative effects of landscape and local factors on the diversity of flower-visitor functional groups in a moderately urbanized city, Wuhan, located in central China. We found that the proportion of impervious surface had no significant effect on the number of visitations, but it was negatively associated with the diversity of flower-visitor groups. The number of visitations by Halictidae and Lepidoptera correlated positively with local flower density and flowering plant species richness, respectively. Flowering plant species richness was also positively correlated with the diversity of flower-visitor groups. The proportion of green space was negatively associated with the visitation number of Muscidae and the overall diversity of flower-visitor groups, revealing the potential influence of green space quality on pollinator assemblage. The pollination networks under three urbanization levels (with a total of 11 flower visitor groups and 43 plant species) were asymmetric, highly nested, and generalized. The suburb sites contained the highest diversity of interactions. Core flowering plants (*Oenothera speciosa*, *Coreopsis grandiflora* and *Cyanus segetum*) are exotic species with attractive flowers. Improving green space quality (high flower density and flowering plant species richness) and using attractive native flowering plants (*Nandina domestica*, *Rosa chinensis*, *Astragalus sinicus*, *Cirsium arvense* var. *integrifolium*, and *Zabelia biflora*) would enhance the function of urban green space to maintain pollinator diversity and ecosystem stability.

**Keywords:** flower-visitor; insect pollination; plant species; urbanization; diversity; functional group; pollination network; green space

## 1. Introduction

The sustainable development of human societies relies on ecosystem services provided by nature, among which pollination service is vital and vulnerable [1]. Pollinators, such as bees, butterflies, and hoverflies have declined globally [2–6]. Flying insects in the nature reserves of Germany have declined by more than 75% during 1989–2016 [7]. The relative

abundance of four bumblebee species in North America has decreased by 96% from 1980s to 2009 [8]. The primary factors causing large-scale pollinator decline include habitat loss and fragmentation, agricultural intensification, use of agricultural chemicals, pathogen infestation, invasive alien species, climate change, light pollution, and their collective interactions [8–11].

Urbanization is a dynamic process involving dramatic and continuous changes in land use, including a decrease in bare land, an increase in the area covered by impervious surfaces, and the loss of natural vegetation [10,12,13]. In general, urbanization causes habitat loss and reduces the habitat suitability of pollinators, and consequently reduces the abundance and diversity of pollinators [14–17]. However, some studies have found that moderate urban expansion or artificial land-use may increase the number of certain pollinators and their pollination services. Landscape heterogeneity is improved at the landscape level, resulting in diverse habitat types for different pollinator groups [18]. On the other hand, urban green spaces can provide suitable alternative habitats and food sources for pollinators [19,20]. Some studies have shown that floricultural suburban gardens contribute more to bumblebee population growth and nesting than agro-ecosystems or other rural landscapes [21–23]. Gardens with high flower diversity and density are considered to be pollinator-friendly [24]. To increase the local nectar resources, the cutting frequency of the green space should be reduced, in order to retain herb and shrub layers [25]. Therefore, suitable urbanized areas may play an important role in pollinator conservation.

At the landscape scale, residual semi-natural habitats in urban environments can serve as habitats and shelters for pollinators [26]. The impact of urban environments on insects is largely determined by the number and distribution of semi-natural habitats such as green spaces, lawns, hedges, and so forth [25]. Urban night light may directly affect the nocturnal pollinators and pollination networks, and indirectly affect the diurnal pollinators. Artificial night light disturbs nocturnal pollination and reduces plant reproduction, which also provide food resources for diurnal pollinators [27]. On the other hand, the night light may decrease nectar depletion by nocturnal pollinators and favor foraging by diurnal pollinators [28]. At the local scale, vegetation type and abundance of flower resources affects the habitat suitability and food availability for pollinators. For example, a high abundance and species richness of flowering plants can provide nectar and other food resources in different seasons, which can help maintain high pollinator diversity [20,29]. The plant species differ markedly in their attractiveness to pollinators [30,31], and species of *Origanum*, *Agastache*, *Lavandula*, and *Nepeta* were reported to be highly attractive [30]. Pollinators differ in their preference of floral traits [32], so the species of flowering plants also affects the pollinator assemblage. Therefore, studies of both landscape factors and local factors in moderately urbanized areas can help reveal the impact of urbanization on pollinator diversity and provide guidance in the planning of urban green spaces.

In this study, we investigated the effects of landscape factors and local factors driven by urbanization on the diversity of flower visitors in urban and suburban green spaces in Wuhan, central China. The objectives of this study were: (1) to investigate the effects of landscape factors (percentage of impervious surface, percentage of green space, and night light intensity) and local factors (species richness of flowering plants, and flower density) on the abundance and diversity of flower-visitor groups; (2) to investigate the effects of urbanization level on interaction between plants and flower-visitor groups, and clarify the key plant species that may play an important role in maintaining urban pollinator diversity.

## 2. Materials and Methods

### 2.1. Study Area

The study system we surveyed was located in Wuhan City (113°41′–115°05′ E, 29°58′–31°22′ N, ranging between 19.2 and 873.7 m in elevation and with a total area of 856,915 hectares) in Hubei Province, central China. This is one of the most rapidly growing cities in central China, where urban and arable areas have largely expanded over natural vegetation.

According to Wuhan's General Plan for Land Use (2006–2020), construction land was expected to reach 185,000 ha, accounting for 21.58% of total land area in 2020 [33].

Field investigations were conducted from May to July in 2019 at 19 independent sites in urban parks, covering an urbanization gradient. The study sites were selected on the basis of the proportion of impervious surfaces around the park within a 2000 m radius (Table A1, Figure 1), using Google satellite imagery 2016 and Hubei land cover maps (resolution of 30 m) [34]. We used 2000 m as the research scale of landscape variables for the following reasons. Firstly, some flower-visitors are capable of flying long distances, for example, the foraging flight of bumblebee workers is greater than 1 km and their maximum measurement reaches approximately 2 km [35]. Consequently, the habitat and resource that pollinators can reach cannot be evaluated at a small scale, and the effects of impervious space on the pollinator abundance are more apparent on a larger scale [36]. Secondly, the area of urban parks usually exceeds 100 hectares (for example, the area of Shizishan Parkland is 495 hectares, and the area of Wuhan Garden Expo Park is 213.8 hectares). We excluded the park area in calculation of impervious surface, because the parks with large squares and pavements usually lead to overestimation of urbanization level in their settings, especially for exurb parks. The water cover was excluded in the calculation of land cover composition following the approach used by Xie et al. (2017) [37]. Study sites were situated at least 3 km from each other. In keeping with McKinney's categorization of urban landscapes [38], the "City core" represented the area in which the impervious surfaces were greater than 80%, in the "Suburb" the impervious area was between 50 and 80%, and the impervious surface cover of the "Exurb" comprised sites with less than 50% coverage.

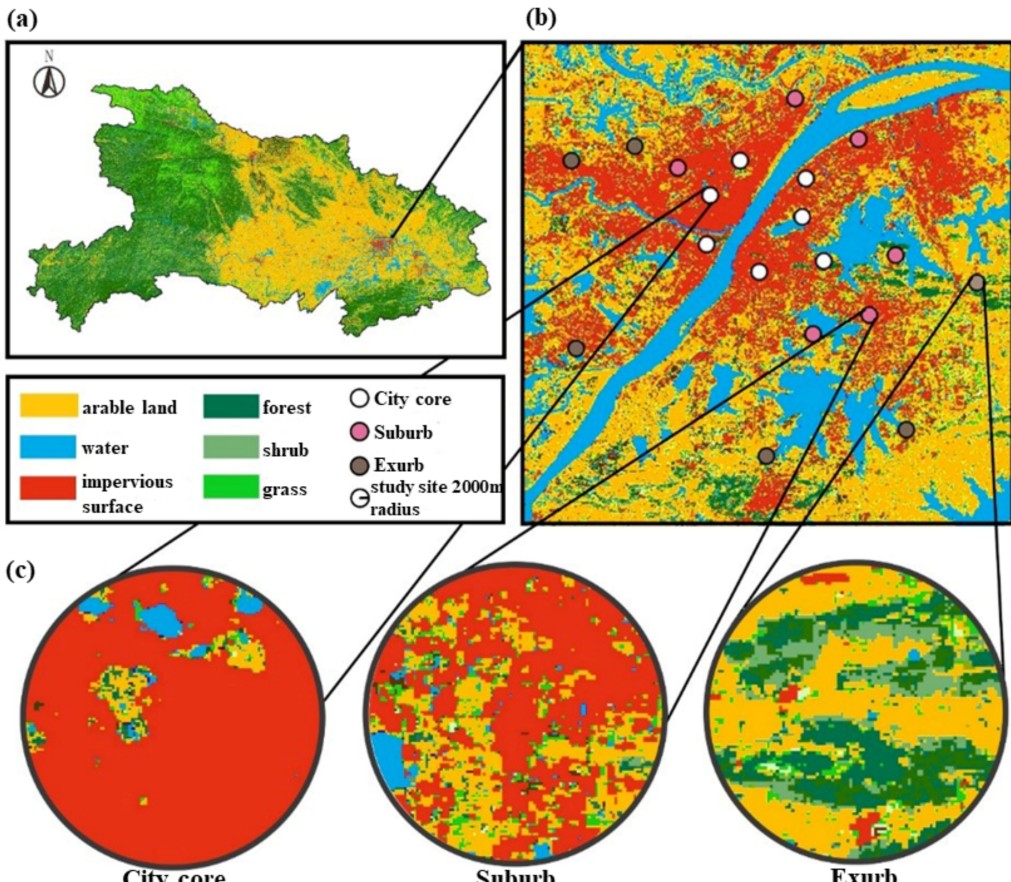

**Figure 1.** Location of the study sites. (**a**) The location of Wuhan City in Hubei Province, central China. (**b**) The location of 19 study sites in Wuhan. (**c**) The categories of urbanization level based on the proportion of impervious surface.

## 2.2. Landscape Variables

We calculated the proportion of green space within a 2000 m radius (excluding areas covered by water) around the study sites (Tables 1 and A1). Green space included parklands, protective green space, affiliated green space, regional green space, and other categories of non-developed settings.

**Table 1.** Description of landscape and local factors.

| Factor | Scale | Description | Level |
|---|---|---|---|
| Proportion of impervious surface | r = 2000 m | Proportion of impervious surface around the park, as an indicator of urbanization level | Continuous variable, 0–100% |
| Proportion of green space | r = 2000 m | Proportion of green space around the study site, including parkland, protective green space, affiliated green space, regional green space, and other types | Continuous variable, 0–100% |
| Night light intensity | r = 2000 m | Average brightness of night light around the sites | Continuous variable |
| Density of flowers | 20 m × 20 m plot | Number of flowers per m$^2$ | Continuous variable |
| Species richness of flowering plant | 20 m × 20 m plot | Number of flowering plant species | Discrete variable |
| Sampling time | | The month of field survey | Nominal variable, May, June, July |

The night light data were obtained from remote sensing images by the Luojia No. 1 night remote sensing satellite and have a resolution of 130 m. The average night light intensity within a 2000 m radius around the sites was calculated using Arcmap software (Version:10.6.1) (Tables 1 and A1).

## 2.3. Local Floral Resources and Flower-Visitors

At each site we selected a representative 20 × 20 m plot in which the floral resources were relatively abundant, and the level of human disturbance was low or absent. We conducted surveys between 1000 and 1700 h on sunny and calm days. Three survey rounds were performed in each site once a month.

Before our observations of flower-visitors, we recorded flowering plant species richness and the density of flowers (the number of flowers per meter squared) in each plot. The density of flowers was sampled in 10 randomly selected 1 × 1 m plots. The number of flowers were counted according to the procedure described by Gong and Huang (2009) [32]. Single flower plants (e.g., *Oenothera speciosa*) and plants with clustered florets in their inflorescence (e.g., *Trifolium repens* and composites) were counted as single floral units. For inflorescences with wide and sparse florets that can be visited independently by pollinators (e.g., *Althaea rosea* and *Oxalis articulate*), each floret was counted as a floral unit. Flowering trees higher than 2 m were not included because of the difficulty in surveying their flowers. The anemophilous flowers were not included.

In the 20 × 20 m plots, we recorded flower-visitors and visited plant species using line-transect methodology. We counted and recorded visits to flowers within 1 m of each side of the transect and 1 m in front of the observer at a steady walking speed. For each survey round, the transect was positioned across the 20 × 20 m plot and proceeded for 2 km (Figure 2). Due to the difficulty of species identification in the field, the flower-visitors were identified to genus (*Apis* spp., *Anthophora* spp.), family (Halictidae, Vespidae, Syrphidae), or order level (Lepidoptera, Coleoptera). To avoid interference with insect visitations, we caught individuals for further identification using a hand net as the survey round was completed. The flower visitors were transferred to separate vials and stored at −80 °C. Further identification was conducted utilizing DNA sequence comparisons

(COI-KC and ITS2) based upon insect leg tissue samples (Supplementary Material Table S1). The insect specimens were deposited in the lab of College of Horticulture & Forestry Sciences, Huazhong Agricultural University. The flower-visitors were classified into 11 functional groups: honeybee (*Apis cerana*, *A. dorsata*, *A. mellifera*, *A. nuluensis*), *Anthophora* spp., common wasp (Vespidae and Ichneumonidae), Halictidae (*Lasioglossum seillean*), Muscidae and Tachinidae (*Musca domestica*, *Drino inconspicua*), Lepidoptera (*Euxoa castanea*, *Gonepteryx cleopatra*, *Graphium sarpedon*, *Parnara batta*, *Pseudozizeeria maha*, etc.), Orthoptera (Oedipodidae), Coleoptera (*Dytiscus semisulcatus*), Odonata, ant (Formicidae), and Syrphidae (*Episyrphus balteatus*, *Eristalis tenax*, *Sphaerophoria philanthus*) (Supplementary Material Table S1). The functional groups were classified according to the insect body size and visiting behavior, for example, the short-tongued honeybee, long-tongued and large-sized bee (*Anthophora* spp.), solitary bee (Halictidae), long-tongued butterfly, and moth. We used Shannon's diversity index (*H*) to estimate the diversity of flower-visitor groups for each sample plot during each survey.

$$H = -\sum(Pi) \times (\ln Pi)$$

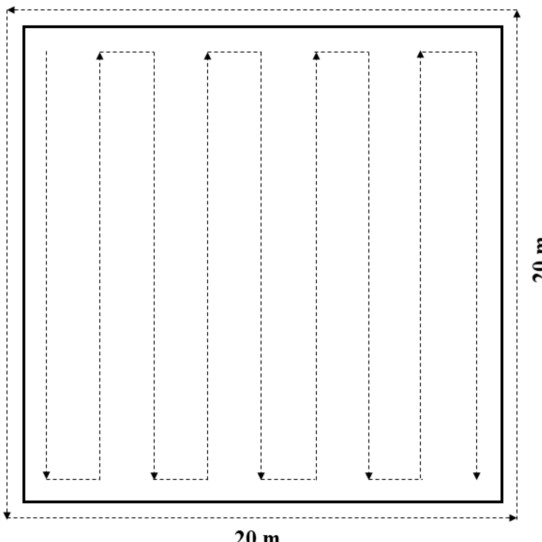

**Figure 2.** Transect route of flower-visitor survey.

*Pi*: the percentage of visits by each flower-visitor group to total visits.

Due to revegetation and plant replacement in green space areas, there were four times when there were no flowering plants present in the sample sites (Shouyi Cultural Park and Simeitang Park in the second round, Shizishan Parkland and Dijiao Park in the third round). The sites without flowering plants in the survey round were excluded from analysis.

*2.4. Data Analysis*

We examined the effect of the proportion of impervious surface, proportion of green space, night light intensity, local flower density, flowering plant species richness, and the sampling time (May, June, and July) on the number of visits by flower-visitors and the diversity of flower-visitor groups during each survey round, using the generalized linear mixed model (GLMM). The study site was included as a random effect. Variance inflation factors (VIFs) were checked to ensure that the factors met the assumption of independence. None of the factors had VIFs greater than 4, so the factors could be included independently in the model [31].

The number of visits by flower-visitor groups (discrete data with a large variance) were analyzed with a negative binomial distribution (with log link) [39]. We examined the number of visits by all groups and also by the main visitor groups (honeybee, Halictidae,

Muscidae, Lepidoptera). Honeybees are extremely abundant and widely distributed, and may drive the results. To address this possibility, we also analyzed the number of visits with honeybees excluded. The diversity of flower-visitor functional groups was analyzed with a normal distribution.

To illustrate the effect of urbanization level on the interaction between flowering plants and flower-visitor groups, we applied the Shannon's diversity index of interactions (*Pi* was calculated as the proportion of interactions) for each urbanization level. We also investigated the pollination networks for the three urbanization levels using R software (Version:4.0.3, 'bipartite' package).

## 3. Results

### 3.1. Survey Results

In total, we recorded 4115 floral visits on 53 plant species. The honeybees made 2534 visits and accounted for 61.58% of the total visitation, followed by Halictidae (625 visits, 15.19%), Lepidoptera (449 visits, 10.91%), and Muscidae and Tachinidae (335 visits, 8.14%). Other groups, including Formicidae (92 visits), *Anthophora* spp. (35 visits), Coleoptera (17 visits), Syrphidae (15 visits), Vespidae and Ichneumonidae (6 visits), Odonata (5 visits), and Orthoptera (2 visits), accounted for less than 5% of the total visits.

The study sites supported a diverse array of flowering plant species. We observed 53 plant species across 28 plant families. The most common plant families were the Asteraceae (13 species) and Leguminosae (4 species). Thirty-one plant species were exotic, of which nine species were invasive, including *Trifolium repens*, *Erigeron annuus*, *Trifolium pratense*, *Oenothera biennis*, *Veronica persica*, *Daucus carota*, *Geranium carolinianum*, *Silybum marianum*, and *Zephyranthes carinata* (Table A2) [40,41]. Each site contained a mean of 2.84 (±2.59) species of flowering plant during each round. The density of flowers ranged from 0.4 to 535.6 flower/m$^2$. The most common flowering plant species was *Rosa chinensis*, which was found in six sites, followed by *Oenothera speciosa* (five sites), *Erigeron annuus* (five sites), *Trifolium repens* L. (four sites), *Coreopsis grandiflora* (four sites), *Cosmos bipinnatus* (four sites), *Oxalis articulata* (four sites), and *Veronica persica* (four sites) (Table A2).

In this survey, a few plant species received the most visits (such as *Oenothera speciosa*, *Coreopsis grandiflora*, *Centaurea cyanus*, and *Oxalis articulata*). However, no visitor groups were observed for 10 plant species, *Weigela florida*, *Plantago depressa*, *Geranium carolinianum*, *Salvia miltiorrhiza*, *Silybum marianum*, *Oxalis corniculate*, *Ophiopogon japonicus*, *Platycodon grandiflorus*, *Canna indica*, and *Zephyranthes carinata*. These species were rare, found in only one survey site and in one survey round.

### 3.2. Factors Influencing the Number of Visits by Flower-Visitor Groups

The sampling time significantly affected the number of visits by all groups during each survey round (GLMM, $F_{2,45} = 3.365$, $p = 0.043$; Table 2). Visits by all groups were significantly higher in May (133.17 ± 125.00) than in June (54.80 ± 63.58) and July (41.53 ± 52.09) (Figure 3a).

None of the factors affected the number of visits when honeybees were excluded from the analysis (Table 2).

The number of visits by honeybees was significantly affected by the sampling time (GLMM, $F_{2,45} = 7.554$, $p = 0.001$; Table 2). During each survey round, honeybee visitations were significantly higher in May (104.17 ± 117.52) than that in June (26.00 ± 38.75) and July (9.27 ± 14.069). The visitation number did not differ between June and July (Figure 3c).

Visits by Halictidae were positively correlated with the flower density (GLMM, fixed coefficient = 0.040, $F_{1,45} = 47.175$, $p < 0.001$; Table 2).

Visits by Lepidoptera were positively associated with flowering plant species richness (GLMM, fixed coefficient = 0.298, $F_{1,45} = 6.317$, $p = 0.016$; Table 2).

Visits by Muscidae were affected by the sampling time (GLMM, $F_{2,45} = 10.366$, $p < 0.001$; Table 2, Figure 3f). During each round, the Muscidae made significantly more visits in May ($13.56 \pm 19.986$), followed by June ($4.35 \pm 7.46$), making least visits in July ($1.27 \pm 2.915$) (Figure 2). The number of visits by Muscidae was negatively associated with the proportion of green space (GLMM, fixed coefficient = $-5.384$, $F_{1,45} = 4.393$, $p = 0.042$; Table 2).

**Table 2.** Effects of fixed factors on the number of visits by flower-visitor groups (GLMMs).

| Term | Factor | Fixed Coefficient | F | p |
|---|---|---|---|---|
| All flower-visitors | Proportion of impervious surface | −1.579 | 0.666 | 0.419 |
| | Sampling time | — | 3.365 | 0.043 * |
| | Proportion of green space | −2.556 | 0.772 | 0.384 |
| | Night light intensity | <−0.001 | 0.176 | 0.677 |
| | Local flower density | 0.001 | 0.155 | 0.696 |
| | Flowering plant species richness | 0.154 | 2.001 | 0.164 |
| Non-honeybee visitors | Proportion of impervious surface | −2.309 | 1.410 | 0.241 |
| | Sampling time | — | 0.338 | 0.715 |
| | Proportion of green space | −4.499 | 2.372 | 0.131 |
| | Night light intensity | <−0.001 | 0.530 | 0.471 |
| | Local flower density | 0.003 | 1.021 | 0.318 |
| | Flowering plant species richness | 0.186 | 2.913 | 0.095 |
| Honeybee | Proportion of impervious surface | −2.067 | 1.722 | 0.196 |
| | Sampling time | — | 7.554 | 0.001 *** |
| | Proportion of green space | 0.250 | 0.011 | 0.917 |
| | Night light intensity | <0.001 | 0.121 | 0.730 |
| | Local flower density | −0.001 | 0.037 | 0.848 |
| | Flowering plant species richness | 0.119 | 1.601 | 0.212 |
| Halictidae | Proportion of impervious surface | 1.121 | 0.152 | 0.699 |
| | Sampling time | — | 1.288 | 0.286 |
| | Proportion of green space | −1.607 | 0.134 | 0.716 |
| | Night light intensity | <−0.001 | 0.043 | 0.837 |
| | Local flower density | 0.004 | 47.175 | <0.001 *** |
| | Flowering plant species richness | −0.116 | 0.448 | 0.507 |
| Lepidoptera | Proportion of impervious surface | −1.108 | 0.265 | 0.609 |
| | Sampling time | — | 0.525 | 0.595 |
| | Proportion of green space | −3.400 | 1.101 | 0.300 |
| | Night light intensity | <−0.001 | 0.641 | 0.428 |
| | Local flower density | 0.004 | 1.673 | 0.202 |
| | Flowering plant species richness | 0.298 | 6.317 | 0.016 * |
| Muscidae | Proportion of impervious surface | −1.023 | 0.348 | 0.558 |
| | Sampling time | — | 10.366 | <0.001 *** |
| | Proportion of green space | −5.384 | 4.393 | 0.042 * |
| | Night light intensity | <−0.001 | 2.087 | 0.155 |
| | Local flower density | 0.002 | 0.364 | 0.549 |
| | Flowering plant species richness | −0.005 | 0.003 | 0.956 |

Note: * $p < 0.05$, *** $p < 0.01$.

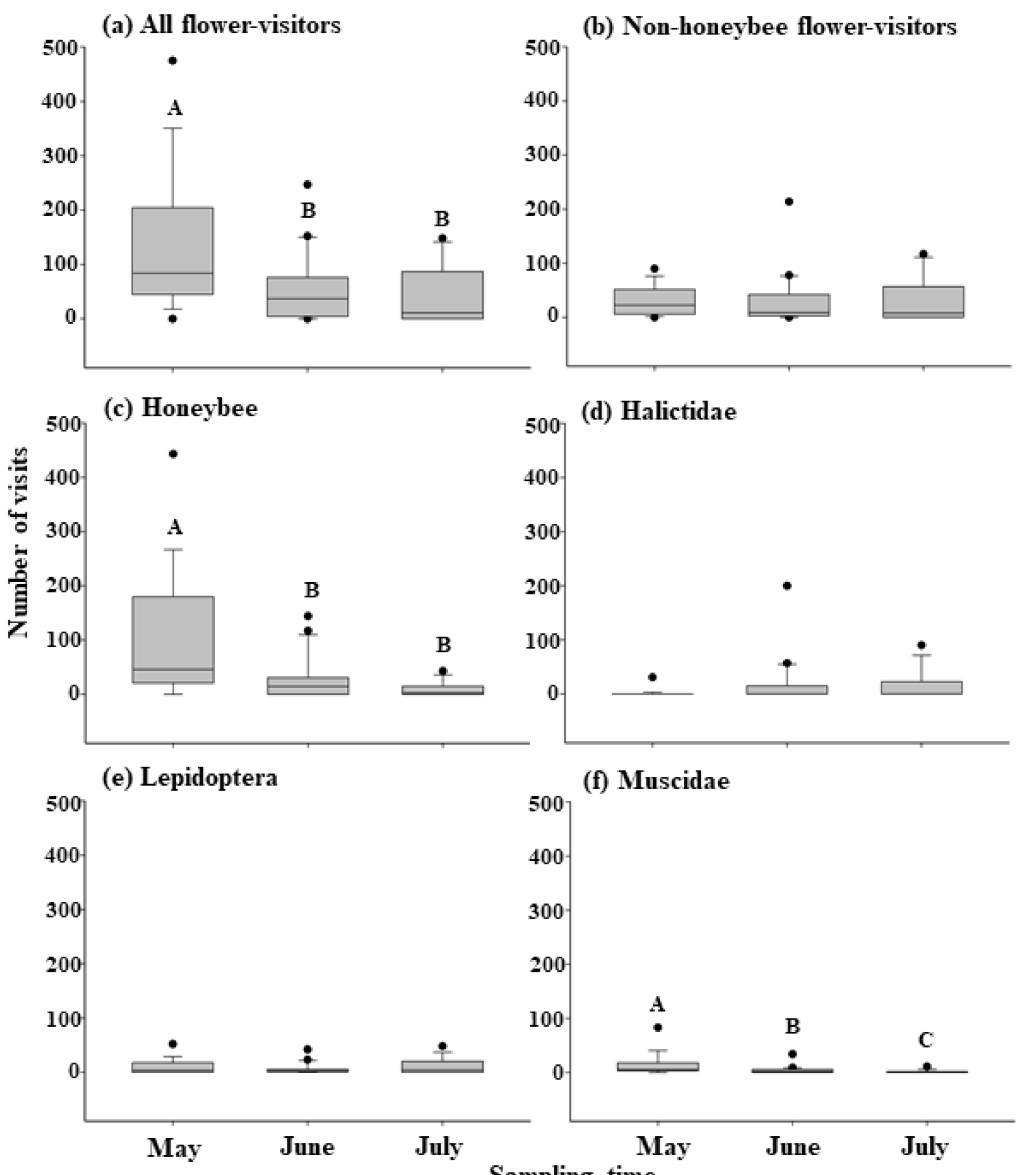

**Figure 3.** The effects of sampling time on the number of visits by (**a**) All flower visitors, (**b**) Non-honeybee visitors, (**c**) Honeybee, (**d**) Halictidae, (**e**) Lepidoptera, (**f**) Muscidae. Boxes with different capital letters indicate significant differences among three sampling months according to GLMMs ($p < 0.05$).

### 3.3. Factors Influencing the Composition and Diversity of Flower-Visitor Groups

There were six, ten, and eight functional groups of flower-visitors recorded in the City core, Suburb, and Exurb sites, respectively. Honeybees, Halictidae, Lepidoptera, Muscidae, and Tachinidae occurred under the three urbanization levels. Orthoptera only occurred in Suburb sites. Vespidae and Ichneumonidae were only found in Exurb sites. The visitor groups that appeared in City core sites were also recorded in the Suburb and Exurb sites.

The diversity of flower-visitor functional groups during each survey was negatively associated with the proportion of impervious surface and proportion of green space (GLMM, proportion of impervious surface, fixed coefficient = −1.179, $F_{1,45} = 4.847$, $p = 0.033$; proportion of green space, fixed coefficient = −1.864, $F_{1,45} = 5.223$, $p = 0.027$; Table 3, Figure 4a,b). The diversity of flower-visitor groups was positively correlated with the species richness of flowering plant (GLMM, fixed coefficient = 0.072, $F_{1,45} = 5.005$, $p = 0.030$; Table 3, Figure 4c).

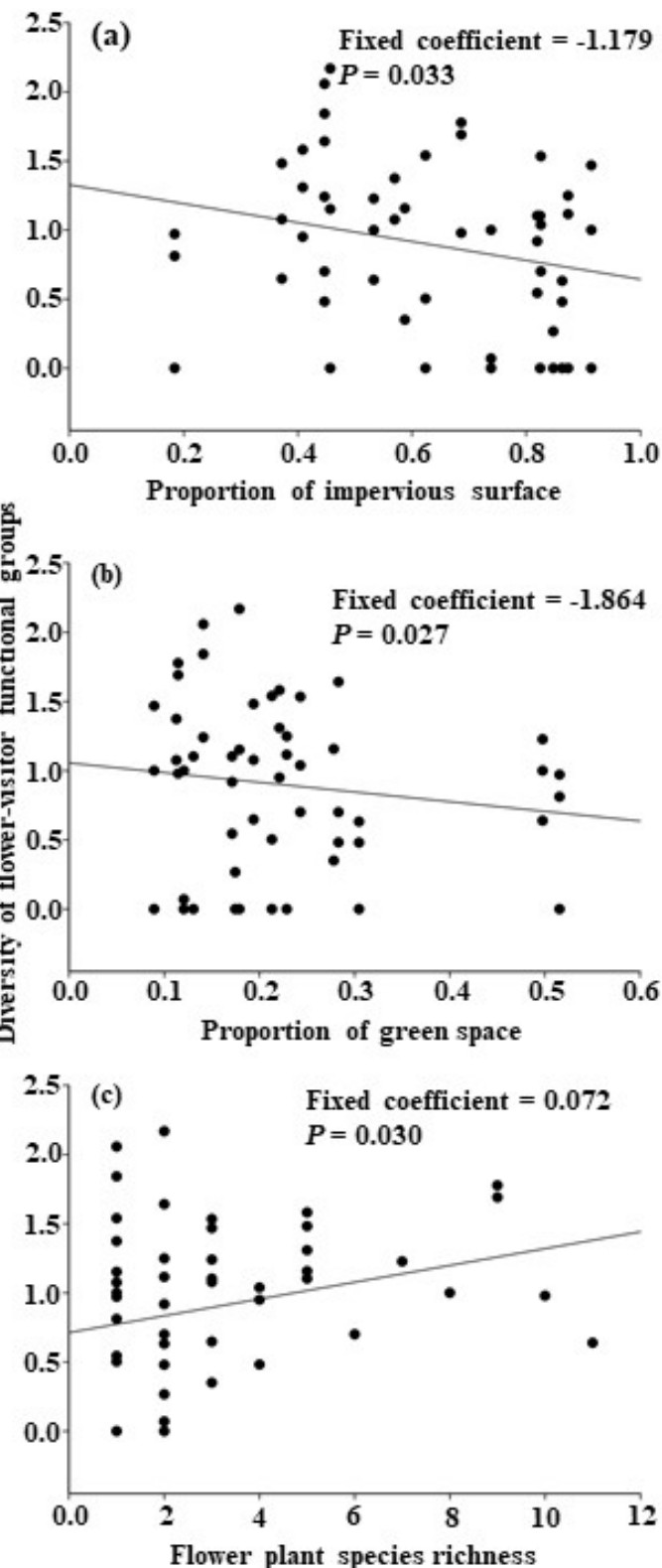

**Figure 4.** The correlation between diversity of flower-visitor functional groups and (**a**) Proportion of impervious surface, (**b**) Proportion of green space, (**c**) Flower plant species richness according to GLMMs.

**Table 3.** Effects of fixed factors on the diversity of flower-visitor functional group (GLMM).

| Factor | Fixed Coefficient | F | $p$ |
|---|---|---|---|
| Proportion of impervious surface | −1.179 | 4.847 | 0.033 * |
| Sampling time | — | 0.172 | 0.842 |
| Proportion of green space | −1.864 | 5.223 | 0.027 * |
| Night light intensity | <0.001 | 0.008 | 0.929 |
| Local flower density | <0.001 | 0.199 | 0.658 |
| Flowering plant species richness | 0.072 | 5.005 | 0.030 * |

Note: * $p < 0.05$, *** $p < 0.01$.

### 3.4. The Interaction between Flowering Plants and Flower-Visitor Groups under Three Urbanization Levels

The Shannon's diversity index results of interactions in the City core, Suburb, and Exurb sites were 2.8235, 2.9426, and 2.5174, respectively. The Suburb sites contained the most flower-visitor groups (10 groups) and flowering plant species (34 species), and it had the highest diversity of interactions.

The pollination networks were constructed for three urbanization levels, with a total of 11 flower-visitor groups (P1–P11) and 43 plant species which received visits (F1~F43) (Figure 5, Tables 4, A2 and A3). For the City core, Suburb, and Exurb sites, the Connectance (calculated as the abundance of links/abundance of potential links) was 0.31, 0.21, and 0.38, respectively, indicating that some potential links may have not been observed, especially at the Suburb sites (Table 4). The nestedness of the pollination network (calculated as (100-T)/100, T referring to the matrix temperature) was 0.89, 0.95, and 0.88 for three urbanization levels, indicating that the pollination networks were highly nested (Table 4). The pollination networks were highly generalized. The most connected visitor groups were honeybees, Lepidoptera and Muscidae associated with 30, 29, and 28 plant species, respectively (Table A3). The most connected plant species were *Coreopsis grandiflora* and *Centaurea cyanus*, which were visited by eight groups. The most connected native plant species was *Nandina domestica*, which received visits by four groups, followed by *Rosa chinensis*, *Astragalus sinicus*, *Cirsium arvense* var. *integrifolium*, *Zabelia biflora*, and *Medicago falcata*, which were visited by three visitor groups (Table 5).

**Table 4.** Metrics of pollination networks of three urbanization levels.

| Metrics | City | Suburb | Exurb | Total |
|---|---|---|---|---|
| Functional groups of flower-visitors | 6 | 10 | 8 | 11 |
| Plant Species | 18 | 34 | 13 | 43 |
| Links | 754 | 1433 | 1928 | 4115 |
| Max links of flower-visitor groups | 462 | 994 | 1078 | 2534 |
| Max links of plants | 177 | 420 | 1009 | 1018 |
| Abundance of links | 39 | 70 | 40 | 117 |
| Abundance of potential links | 126 | 340 | 104 | 473 |
| Connectance | 0.31 | 0.21 | 0.38 | 0.24 |
| Matrix temperature | 10.80 | 5.16 | 12.12 | 6.25 |
| Nestedness | 0.89 | 0.95 | 0.88 | 0.94 |

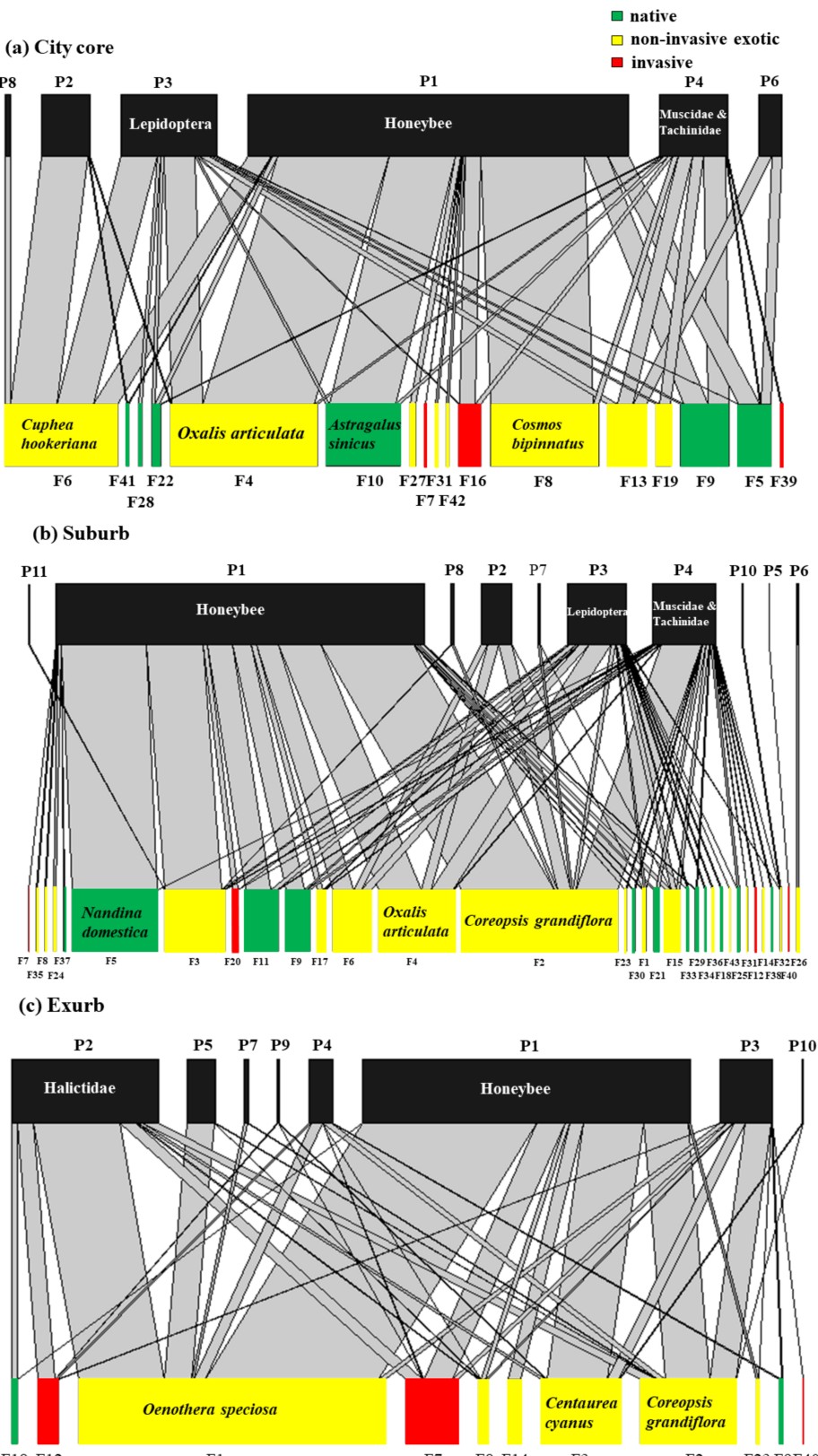

**Figure 5.** Visualization of the pollination networks of (**a**) City core, (**b**) Suburb, and (**c**) Exurb. The width of the lines connecting species is scaled to the number of links. Refer to Tables A2 and A3 for the code of plant species and flower-visitor groups in network.

**Table 5.** Code and links of native plant species in pollination network.

| Code in Network | Plant Species | Number of Links | Number of Linked Pollinator Groups | Normalized Degree |
|---|---|---|---|---|
| F5 | *Nandina domestica* | 269 | 4 | 0.3636 |
| F9 | *Rosa chinensis* | 135 | 3 | 0.2727 |
| F10 | *Astragalus sinicus* | 89 | 3 | 0.2727 |
| F11 | *Cirsium arvense var. integrifolium* | 87 | 3 | 0.2727 |
| F18 | *Zabelia biflora* | 19 | 3 | 0.2727 |
| F21 | *Ligustrum sinense* | 14 | 2 | 0.1818 |
| F22 | *Medicago falcata* | 12 | 3 | 0.2727 |
| F25 | *Sedum lineare* | 7 | 2 | 0.1818 |
| F28 | *Taraxacum mongolicum* | 4 | 1 | 0.0909 |
| F29 | *Dianthus chinensis* | 4 | 2 | 0.1818 |
| F30 | *Alcea rosea Linnaeus* | 4 | 2 | 0.1818 |
| F33 | *Hemerocallis fulva* | 3 | 2 | 0.1818 |
| F34 | *Acorus calamus* | 3 | 1 | 0.0909 |
| F37 | *Orychophragmus violaceus* | 2 | 1 | 0.0909 |
| F38 | *Vaccaria hispanica* | 2 | 1 | 0.0909 |
| F41 | *Kerria japonica* | 2 | 2 | 0.1818 |

## 4. Discussion

### 4.1. Effects of the Proportion of Impervious Surface to Flower-Visitor Groups

The results show that the proportion of impervious surface has no significant effect on the number of visits by flower-visitor groups (Table 2). Studies have reported that pollinators tend to prefer (semi-) natural habitats [42], but the visitor groups respond differently to urbanization. Social bees such as honeybees have different individuals specialized for pollen and nectar collection [43,44], and they are more likely to respond to the different spatial and temporal distribution of flower resources. The solitary bees experience landscapes at small scales [45,46], and may be particularly affected by landscape intensification and changing habitat [47]. The number of visits by other groups (Halictidae, Lepidoptera, Muscidae, Formicidae, etc.) was relatively less. The honeybee was the most abundant, and potentially drove the results.

The composition of flower-visitor groups differed in the three urbanization levels. The number of flower-visitor groups was the least in City core, where Vespidae and Ichneumonidae, Formicidae, Coleoptera, Odonata, and Orthoptera were not recorded. Since Vespidae, Formicidae, and Orthoptera are usually omnipresent and are as abundant in cities as in exurbs, this likely indicates a lack of detection due to low sampling effort in the study sites. The diversity of visitor groups negatively correlated with the proportion of impervious surface (Table 3, Figure 4a). This trend is consistent with previous studies. Abbate et al. (2019) found that species richness and diversity of bees were lower in highly urbanized areas [48]. Burdine and McCluney (2019) determined that bee diversity decreased as the proportion of impervious surface increased [49]. Collado et al. (2019) compared species richness and diversity of bees in natural habitats with those in urban core areas and found that natural habitat was better able to maintain bee diversity [42]. Persson et al. (2020) determined that the species richness of wild bees was decreased in densely built-up areas [50]. Our approach of categorizing flower-visitors to functional groups has limitations in clarifying the effect of urbanization on pollinator species, and thus may hardly be compared with findings of previous studies at species level. More studies at a species level would better clarify how different genera or species react to urbanization.

*4.2. Effects of Proportion of Green Space on Flower-Visitor Groups*

The proportion of green space caused no significant effect on visitation by honeybees, Halictidae, and Lepidoptera, but, contrary to previous studies, it was negatively associated with the number of visits by Muscidae and reduced the diversity of flower-visitor groups (Tables 2 and 3, Figure 4b). Green space is usually considered to be one of the important habitats for pollinators, and a high proportion of green space can significantly increase the number of pollinators [51,52]. However, studies also found that visitation by honeybees did not decrease as the distance between the study site and semi-natural habitat increased [53,54]. This was likely due to the wider foraging range available and a better adaptation to environmental change [53]. Honeybees were abundant and may have driven the results. Additionally, the low detection of visits by Halictidae, Lepidoptera, and other rare visitor groups increases the difficulty of analyzing the effect of the proportion of green space at our sample sites. The visits by Muscidae and the diversity of flower-visitor groups were negatively associated with the proportion of green space. The probable reason is that, in addition to the size or proportion of green space, the quality of green space also has a significant impact on pollinator visitation. Artificial green space in the city mainly exists in the form of park and residential green space, and only functions as a habitat connector or as temporary habitat. By contrast, natural habitat can better accommodate pollinators as their long-term habitat [42]. Various studies have investigated on how to enhance the habitat value of urban green space. First, 'Pollinator friendly gardens', with high flower diversity, abundance, and density, are considered to have a positive impact of maintaining pollinator diversity [24,55]. Second, as the pollinators diverge in active season, the selection of pollinator-attractive flowering plants should consider both the flowering period of plants and the seasonal activity of pollinators [31]. Third, management measures also play important role in pollinator conservation, including reducing the cutting frequency of meadows, retaining wide herbaceous margins and shrub layers with high nectar resources, and avoiding excessive growth in hedgerow height to reduce the barrier effect [25].

*4.3. Effects of Night Light Intensity on Flower-Visitors*

Night light intensity is caused by the level of urbanization and the magnitude of human activities, and it also has a significantly negative impact on nocturnal pollinators such as moths [56]. The artificial night lights reduce nocturnal pollination and the reproduction of plants. Diurnal pollinators also forage on these plants for food resource, so the effect of night light may extend through the plant–pollination network, and the diurnal pollinators are affected in the long run [27]. In this study, the night light intensity had no significant effect on the number of visits or the diversity of flower-visitor groups (Tables 2 and 3). This may be due to that no nocturnal pollinators were included in our study. In addition, the regeneration of flowering plants is frequent in urban parks, so the reproduction of certain plant species can't cause long-term effects on plant–pollinator interactions.

*4.4. Effects of Local Flower Density and Flowering Plant Richness to Flower-Visitor Groups*

Flower density and the species richness of flowering plants generally have positive effects on pollination, such as increasing pollinator visits [57], and diversity [19,58,59]. In this study, flower density positively correlated with the number of visits by Halictidae (Table 2). Flowering plant species richness was positively associated with the number of visits by Lepidoptera and the diversity of flower-visitor functional groups (Tables 2 and 3, Figure 4c). Diverse plant resources provide food and nesting resources for more pollinator groups. Studies have shown that high plant species richness has a positive effect on maintaining butterfly species richness due to their rapid response to changes in flowering plant abundance and species richness [60].

*4.5. Effects of Urbanization Level on Flowering Plant–Visitor Group Interactions and Core Species*

Suburb sites exhibited the highest diversity of interactions. However, the Suburb sites also contained high number of flower-visitor groups and plant species, and because of

this, it is possible that some potential links may have been missed (Table 4). Among the three urbanization levels, the pollination networks were asymmetric, highly nested, and generalized. When compared with natural habitats and arable land, the urban areas are occupied by more generalists, which are supported by a higher number of plant species [61].

Core functional groups of flower-visitors (honeybee, Halictidae, Lepidoptera, and Muscidae) contribute the most links (Table A3). Core flowering plants (*Oenothera speciosa, Coreopsis grandiflora, Centaurea cyanus,* and *Oxalis articulata*) are exotic plant species and occurred at five, four, three, and four sites, respectively (Table A2). In the urban green space, flower-visitors may adapt and forage on exotic flowering plants, which somewhat enhances pollinator diversity by providing nectar or pollen at certain seasons, when native plants are not actively in flowering period [31,62,63]. However, the presence of attractive exotic and horticulturally modified flowering plants may displace endemic plants that pollinators are adapted to, and compete with native plants for pollinators, thus altering the plant–pollinator network [64–66]. The native plant species, *Nandina domestica*, *Rosa chinensis*, *Astragalus sinicus*, *Cirsium arvense* var. *integrifolium*, and *Zabelia biflora*, were relatively attractive to a diversity of visitor groups (Table 5). The use of these native flowering plants can help maintain the stability of flower–pollinator network and mitigate the negative effects of urbanization.

*4.6. Effects of Sampling Month to Flower-Visitor Groups*

The survey covered three months, during which the plant species, richness, and flower density changed with the seasonal climate. There are significant seasonal differences in the activity of various pollinator groups. Maintaining a diverse set of flower-visitor groups can help meet the pollination needs of different flowering plants.

**5. Conclusions**

This study has some limitations: Firstly, the approach of identifying flower-visitors to broad taxonomic levels (mostly genus or family) rather than species was inadequate for clarifying the responses of different flower-visitor species to urbanization. The conservation status of different flower-visitor species was ignored. Secondly, most of the interactions were between the abundant plant species and main flower-visitor groups, which are widely distributed and generalized. The sampling effort was not adequate to assess the flower-visitor assemblage of less abundant plants, as well as the effect of urbanization on rare and specialized flower-visitors. Thirdly, in the field survey of flower-visitors, transects in the plot were close, so the same individual may have been counted twice or more. To ensure data independence, transects should ideally be linear.

This study provides suggestions for the planning of urban green spaces and provides a theoretical reference for local vegetation configurations. Improvement of the quality of green spaces (high plant species richness, and flower density) and the use of attractive native flowering plants (such as *Nandina domestica*, *Rosa chinensis*, *Astragalus sinicus*, *Cirsium arvense* var. *integrifolium*, and *Zabelia biflora*) can help to reduce the negative effects of city expansion on pollinator diversity, and enhancing the function of urban green space in sustaining biodiversity and ecosystem stability.

**Supplementary Materials:** The following supporting information can be downloaded at: https://www.mdpi.com/article/10.3390/d14030208/s1, Table S1: Species list of flower-visitors according to DNA sequence comparisons; Table S2: The visits and diversity of flower-visitors; Table S3: The interaction between plants and flower-visitor groups under three urbanization levels.

**Author Contributions:** Development of the idea, H.W.; data collection, M.H., N.R. and H.J.; data analysis, M.H., Z.H. and Y.D.; writing-original draft preparation, M.H.; writing-review and editing, H.W., X.L. and P.A.B.; supervision of the project, D.X. All authors have read and agreed to the published version of the manuscript.

**Funding:** This research was funded by the National Natural Science Foundation of China, grant number 31600189, 32071683.

**Institutional Review Board Statement:** Not applicable.

**Informed Consent Statement:** Not applicable.

**Data Availability Statement:** Data is contained within the supplementary material.

**Conflicts of Interest:** The authors declare no conflict of interest.

## Appendix A

**Table A1.** Information of Study Sites.

| Study Site | Latitude | Longitude | Proportion of Impervious Surface | Proportion of Greenspace | Average Night Light Intensity (DN) |
|---|---|---|---|---|---|
| City core | | | | | |
| Jiefang Park | 30.6091 | 114.2911 | 87.35% | 22.87% | 45141 |
| Shahu Park | 30.5758 | 114.3416 | 81.95% | 17.13% | 50630 |
| Luojiashan Parkland | 30.5379 | 114.3561 | 86.32% | 30.47% | 30709 |
| Lianhuahu Park | 30.5517 | 114.2741 | 82.58% | 24.29% | 38430 |
| Shouyi Cultural Park | 30.5433 | 114.3006 | 84.75% | 17.45% | 45896 |
| Zhongshan Park | 30.5858 | 114.2674 | 91.42% | 8.93% | 45881 |
| Simeitang Park | 30.5965 | 114.3313 | 82.46% | 13.07% | 47913 |
| Suburb | | | | | |
| Shizishan Parkland | 30.4750 | 114.3380 | 58.76% | 27.81% | 25860 |
| Houxianghe Park | 30.6115 | 114.2482 | 68.63% | 11.45% | 47191 |
| Heping Park | 30.6369 | 114.3814 | 62.38% | 21.31% | 28522 |
| Wuhan Botanical Garden | 30.5483 | 114.4160 | 53.32% | 49.76% | 10606 |
| Dijiao Park | 30.6655 | 114.3331 | 56.96% | 11.27% | 21320 |
| Guanshan Holland Park | 30.4964 | 114.3927 | 73.86% | 12.05% | 27054 |
| Exurb | | | | | |
| Wuhan Garden Expo Park | 30.6209 | 114.2156 | 44.67% | 28.30% | 38582 |
| Shimenfeng Park | 30.5166 | 114.4735 | 18.43% | 51.55% | 16231 |
| Canglongdao Wetland Park | 30.4069 | 114.4188 | 37.20% | 19.38% | 21667 |
| Jiangxia Central Park | 30.3796 | 114.3159 | 44.67% | 14.10% | 24925 |
| Zhuyehai Park | 30.6237 | 114.1607 | 45.68% | 17.89% | 29033 |
| Tanghu Park | 30.4727 | 114.1556 | 40.86% | 22.10% | 22490 |

Note: Proportion of impervious surface was calculated within a 2000 m radius (excluding water cover) around the parks. Proportion of green space was calculated within a 2000 m radius (excluding water cover) around the study sites. DN refers to digital number of pixel brightness.

**Table A2.** List of Surveyed Plant Species.

| Family | Species | Region of Origin | Status | Distribution Sites | Code in Pollination Network |
|---|---|---|---|---|---|
| Amaryllidaceae | *Tulbaghia violacea* Harv. | South Africa | non-invasive exotic | Suburb (Houxianghe Park, Wuhan Botanical Garden) | F36 |
| Amaryllidaceae | *Zephyranthes carinata* Herbert | South America | invasive | Suburb (Wuhan Botanical Garden) | F53 |
| Araceae | *Acorus calamus* L. | China | native | Suburb (Wuhan Botanical Garden) | F34 |
| Asteraceae | *Carthamus tinctorius* L. | Central Asia | non-invasive exotic | Suburb (Wuhan Botanical Garden) | F35 |
| Asteraceae | *Centaurea cyanus* L. | Europe, Russia, north America | non-invasive exotic | Suburb, Exurb (Shizishan Parkland, Wuhan Botanical Garden, Canglongdao Wetland Park) | F3 |
| Asteraceae | *Cirsium arvense* var. *integrifolium* Wimm. & Grab. | East and north Asia | native | Suburb (Wuhan Botanical Garden) | F11 |
| Asteraceae | *Coreopsis grandiflora* Hogg ex Sw. | America | non-invasive exotic | Suburb, Exurb (Houxianghe Park, Dijiao Park, Jiangxia Central Park, Tanghu Park) | F2 |
| Asteraceae | *Cosmos bipinnatus* Cav. | Mexico | non-invasive exotic | City, Suburb, Exurb (Shouyi Cultural Park, Shizishan Parkland, Houxianghe Park, Wuhan Garden Expo Park) | F8 |
| Asteraceae | *Erigeron annuus* (L.) Pers. | North America | invasive | City, Suburb, Exurb (Jiefang Park, Houxianghe Park, Canglongdao Wetland Park, Jiangxia Central Park, Tanghu Park) | F12 |
| Asteraceae | *Euryops pectinatus* (L.) Cass. | South Africa | non-invasive exotic | Suburb (Houxianghe Park) | F17 |
| Asteraceae | *Helianthus annuus* L. | North America | non-invasive exotic | Suburb (Houxianghe Park) | F26 |
| Asteraceae | *Leucanthemum vulgare* Lam. | West Europe | non-invasive exotic | Suburb, Exurb (Houxianghe Park, Jiangxia Central Park) | F15 |
| Asteraceae | *Sanvitalia procumbens* Lam. | Mexico | non-invasive exotic | Suburb, Exurb (Shizishan Parkland, Canglongdao Wetland Park) | F23 |
| Asteraceae | *Silybum marianum* (L.) Gaertn. | Europe, Mediterranean, north Africa, central Asia | invasive | Suburb (Wuhan Botanical Garden) | F48 |
| Asteraceae | *Tagetes erecta* L. | Mexico | non-invasive exotic | City, Suburb (Shouyi Cultural Park, Houxianghe Park) | F31 |
| Asteraceae | *Taraxacum mongolicum* Hand.-Mazz. | China, north Korea, Mongolia, Russia | native | City (Shahu Park, Lianhuahu Park) | F28 |
| Berberidaceae | *Nandina domestica* Thunb. | China | native | City, Suburb (Jiefang Park, Lianhuahu Park, Guanshan Holland Park) | F5 |

**Table A2.** *Cont.*

| Family | Species | Region of Origin | Status | Distribution Sites | Code in Pollination Network |
|--------|---------|------------------|--------|--------------------|----------------------------|
| Campanulaceae | *Platycodon grandiflorus* (Jacq.) A. DC. | East and north Asia | native | Suburb (Wuhan Botanical Garden) | F51 |
| Cannaceae | *Canna indica* L. | Japan | non-invasive exotic | Suburb (Wuhan Botanical Garden) | F52 |
| Caprifoliaceae | *Weigela florida* (Bunge) A. DC. | China, japan, india, america | native | City (Shahu Park) | F44 |
| Caprifoliaceae | *Zabelia biflora* (Turcz.) Makino | China, korea | native | Suburb, Exurb (Wuhan Botanical Garden, Tanghu Park) | F18 |
| Caryophyllaceae | *Dianthus chinensis* L. | China | native | Suburb (Houxianghe Park) | F29 |
| Caryophyllaceae | *Vaccaria hispanica* (Miller) Rauschert | Europe, asia | native | Suburb (Shizishan Parkland) | F38 |
| Crassulaceae | *Sedum lineare* Thunb. | China, japan, vietnam | native | Suburb (Houxianghe Park) | F25 |
| Cruciferae | *Orychophragmus violaceus* (Linnaeus) O. E. Schulz | China, north Korea | native | Suburb (Shizishan Parkland) | F37 |
| Geraniaceae | *Geranium carolinianum* L. | America | invasive | City (Lianhuahu Park) | F46 |
| Lamiaceae | *Salvia miltiorrhiza* Bunge | China, japan | native | Suburb (Wuhan Botanical Garden) | F47 |
| Lamiaceae | *Scutellaria barbata* D. Don | India, nepal, myanmar, laos, thailand | non-invasive exotic | City (Shouyi Cultural Park) | F42 |
| Leguminosae | *Astragalus sinicus* L. | China | native | City (Luojiashan Parkland) | F10 |
| Leguminosae | *Medicago falcata* L. | China | native | City (Shahu Park, Luojiashan Parkland, Lianhuahu Park) | F22 |
| Leguminosae | *Trifolium pratense* L. | Central Europe | invasive | City (Shahu Park) | F16 |
| Leguminosae | *Trifolium repens* L. | Europe, north Africa | invasive | City, Suburb, Exurb (Simeitang Park, Houxianghe Park, Zhuyehai Park, Tanghu Park) | F7 |
| Liliaceae | *Hemerocallis fulva* (L.) L. | China, south Europe | native | Suburb (Wuhan Botanical Garden) | F33 |
| Liliaceae | *Ophiopogon japonicus* (L. f.) Ker-Gawl. | China, japan, vietnam, india | native | City (Shahu Park) | F50 |
| Lythraceae | *Cuphea hookeriana* Walp. | Mexico | non-invasive exotic | City, Suburb (Lianhuahu Park, Houxianghe Park) | F6 |
| Malvaceae | *Alcea rosea* Linnaeus | China | native | Suburb (Wuhan Botanical Garden) | F30 |
| Oleaceae | *Ligustrum sinense* Lour. | China | native | Suburb (Houxianghe Park) | F21 |
| Onagraceae | *Oenothera biennis* L. | North America | invasive | Suburb (Wuhan Botanical Garden) | F20 |

| Family | Species | Region of Origin | Status | Distribution Sites | Code in Pollination Network |
|---|---|---|---|---|---|
| Onagraceae | *Oenothera speciosa* Nutt. | America | non-invasive exotic | Suburb, Exurb (Shizishan Parkland, Wuhan Botanical Garden, Wuhan Garden Expo Park, Zhuyehai Park, Tanghu Park) | F1 |
| Oxalidaceae | *Oxalis articulata* Savigny | South America | non-invasive exotic | City, Suburb (Shahu Park, Lianhuahu Park, Simeitang Park, Wuhan Botanical Garden | F4 |
| Oxalidaceae | *Oxalis corniculata* L. | Temperate and subtropical Asia, Europe, Mediterranean, north America | native | City (Lianhuahu Park) | F49 |
| Papaveraceae | *Papaver rhoeas* L. | Europe | non-invasive exotic | Suburb, Exurb (Shizishan Parkland, Wuhan Garden Expo Park, Canglongdao Wetland Park) | F14 |
| Plantaginaceae | *Plantago depressa* Willd. | China, north Korea, Russia, Kazakhstan | native | Exurb (Canglongdao Wetland Park) | F45 |
| Plantaginaceae | *Veronica persica* Poir. | Western Asia, Europe | invasive | City, Suburb, Exurb (Lianhuahu Park, Simeitang Park, Wuhan Botanical Garden, Wuhan Garden Expo Park) | F39 |
| Portulacaceae | *Portulaca grandiflora* Hook. | Brazil | non-invasive exotic | City (Shouyi Cultural Park) | F27 |
| Rosaceae | *Kerria japonica* (L.) DC. | China, japan | native | City (Lianhuahu Park) | F41 |
| Rosaceae | *Rosa chinensis* Jacq. | China | native | City, Suburb, Exurb (Jiefang Park, Zhongshan Park, Houxianghe Park, Heping Park, Guanshan Holland Park, Shimenfeng Park) | F9 |
| Saxifragaceae | *Heuchera micrantha* Douglas | Cenral america | non-invasive exotic | Suburb (Houxianghe Park) | F43 |
| Solanaceae | *Petunia* × *atkinsiana* D. Don ex Loudon | South America | non-invasive exotic | City (Zhongshan Park | F19 |
| Umbelliferae | *Coriandrum sativum* L. | Mediterranean | non-invasive exotic | Suburb (Wuhan Botanical Garden) | F24 |
| Umbelliferae | *Daucus carota* L. | Europe | invasive | Suburb, Exurb (Wuhan Botanical Garden, Canglongdao Wetland Park) | F40 |
| Verbenaceae | *Glandularia* × *hybrida* (Groenland & Rümpler) G. L.Nesom & Pruski | Panama, honduras, venezuela | non-invasive exotic | Suburb (Houxianghe Park) | F32 |
| Violaceae | *Viola cornuta* Desf. | Europe | non-invasive exotic | City (Zhongshan Park) | F13 |

**Table A3.** Code and Links of Flower-Visitor Functional Groups in Pollination Network.

| Code in Network | Functional Groups | Number of Links | Number of Linked Plant Species | Normalised Degree |
| --- | --- | --- | --- | --- |
| P1 | Honeybee | 2534 | 30 | 0.6977 |
| P2 | Halictidae | 625 | 10 | 0.2326 |
| P3 | Lepidoptera | 449 | 29 | 0.6744 |
| P4 | Muscidae and Tachinidae | 335 | 28 | 0.6512 |
| P5 | Formicidae | 92 | 4 | 0.0930 |
| P6 | *Anthophora* spp. | 35 | 3 | 0.0698 |
| P7 | Coleoptera | 17 | 3 | 0.0698 |
| P8 | Syrphidae | 15 | 3 | 0.0698 |
| P9 | Vespidae and Ichneumonidae | 6 | 3 | 0.0698 |
| P10 | Odonata | 5 | 3 | 0.0698 |
| P11 | Orthoptera | 2 | 1 | 0.0233 |

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
