# Peer review of "Effects of Landscape and Local Factors on the Diversity of Flower-Visitor Groups under an Urbanization Gradient, a Case Study in Wuhan, China"

_diversity, doi:10.3390/d14030208_

Round 1

Reviewer 1 Report

Dear Authors,

The manuscript by He et al., entitled “Effects of Landscape and Local Factors on the Diversity of Flower-Visitors under an Urbanization Gradient”, aims to investigate the effects of landscape and local factors on the abundance and diversity of flower-visitors, and analyze the effects of plant species on flower-visitor assemblages.

Overall, the manuscript has the merit of providing information on the drivers of pollinator group diversity and visitation along an urbanization gradient in Wuhan and identifying the most visited plants. The manuscript is well-structured, cites some of the most relevant references on this subject, and is relatively well-written (presents some vocabulary/grammar incorrections throughout the text and some poor sentences). I made some corrections/suggestions to the text on the attached pdf file, but I think that the manuscript will benefit from a revision of the English.

Also, I think that the sampling design and pollinator identification need to be clarified, additional data analysis should be made, and some conclusions need to be rephrased since they are not supported by the data. The poor taxonomic resolution of pollinator data puts some limitations for data interpretation and the uncritical use of higher taxonomic levels may have led to unfounded conclusions that may not be supported if pollinator species are identified and their colonization status (native vs exotic) is valued for biodiversity conservation recommendations.

Here follows my main concerns. Additional comments are made on the attached pdf.

Abstract

Abstract is OK, but the final recommendation is problematic since the authors have not identified pollinator species, only pollinator groups. Some of these groups are species rich while others are not; some may include serious exotic/invasive species and/or species of conservation concern. Since the authors ignored the pollinator species and their conservation status in this study, it is tricky to make recommendations on green space planning in cities to favor pollinator conservation. Do you think that cities will benefit from the plantation of exotic species (like the ones you mentioned) instead of native plants? What if these exotic species are favoring mostly exotic/invasive pollinators?

Line 24 - You may remove the sentence “The night-light reduced the visitation by Halictidae. Flowering” since you do not have evidence of a direct effect. A positive correlation does not mean a causal effect and the driving effect on visitation may be other (e.g. urbanization, human disturbance).

Introduction

The introduction is OK, but may benefit from presenting other studies with recommendations for green space planning and management in cities. Also, information on pollination and on pollinator-friendly native plants shoulsd also be introduced.

Materials and Methods

P99 The legend of Figure 1 is incomplete.

Line 101 - why have you selected a radius of 2km to assess the effects of landscape factors on local pollinators? Please provide a justification.

Why does the sum of “Proportion of impervious surface” and “Proportion of greenspace” may exceed 100% land cover (Appendix A)?

Table 1 - review the types of variables used

The sampling design needs clarification:

P128 – “For each plot, the transect accumulated 2 km.” Instead, you should indicate the transect length in each survey and how it was distributed in the 20mx20m plot.

P128 – “We recorded the species of flower-visitor.” Where is this data? You don´t seem to have identified any pollinator species! Where are the pollinator specimens deposited? What references were used for the identification of pollinator species/groups?

It is also strange in your results and needs clarification:

Among dipterans, you only identified muscids and syrphids, but many other groups are commonly found visiting flowers (tachinids, sarcophagids, calliphorids, fannids, anthomyids, sciarids, phorids, etc). Have you included these flies in “muscids”? Please explain.

Among Apoidea, you only identified honeybees, Anthophora spp and halictids, but other groups are commonly found visiting flowers (andrenids, colletids, megachilids, etc). Where have you included these bees? Please explain.

Also, is there just one Apis in your samples (Apis cerana?) or you do not know that?

The authors should present the sampling completeness of their study. If sampling completeness is poor the interpretation of the results should be made with caution.

Since a major goal of the manuscript is to assess the effects of urbanization on pollinators and pollination, it will be important and more interesting to present the pollination networks for the 3 urbanization levels and provide a comparison of the network metrics between urban, suburban and exurban networks. These networks and their comparison will be more informative than the general network (figure 4), allowing to disentangle the effects of urbanization levels on species interactions.

Results

It is not surprising that urbanization level had no effect on pollinator diversity since the analysis was made at higher taxonomic levels (order, family) and pollinator groups are species rich and widespread. Only by analyzing species-level interactions, the authors may found a turnover in species (within groups) or density compensation effects between urbanization levels and eventually identify more specialized plant-pollinator associations.

Line 222 – The authors state that “Orthoptera only occurred in Suburb sites. Vespidae and Formicidae only occurred in Exurb sites.” Since these insect groups are usually omnipresent and, particularly Formicidae, are abundant both in cities as in exurbs, could the lack of detection be due to low sampling effort in the study sites?  

Line 237 – What are the reasons for the lack of detection of flower visitors in 10 plant species? Are these species rare? Was the sampling effort not adequate to assess the flower visitors of low abundant plants? Please indicate them in the manuscript/supplemental material.

The information on plant-insect interactions can be greatly improved:

The authors should present the pollination networks for the three urbanization levels; pollination network metrics should be compared between urbanization levels; the information on native vs exotic plants should be included in the analysis and figures (a different color for each plant category).

Tables 3, 4 and 6 should be included as supplemental material.

The authors should indicate if the data of this study is available and where can be consulted.

Discussion

The discussion is well-structured, easy to read and many important references were included by the authors. Nevertheless, the authors need to acknowledge that their approach of identifying just pollinator groups (not pollinator species!) has some limitations and thus may hardly be compared with findings at species level by other authors (references in lines 276-283). Also, in some sections the results were again repeated with detail and this should be avoided.

Line 275 - “the wasp, ant, Coleoptera, Odonata and Orthoptera was not recorded in City core”. It is true, but these groups are also common in cities. Any reason for not finding them?

Line 291 – “The higher richness or diversity of flowering plants in the city core may reduce the negative effects of urbanization on flower-visitor diversity.” Can you test this hypothesis in your study and present the results?

Line 304 – You found that “the proportion of green space had a negative effect on the number of visits by Halictidae and Muscidae, which was contrary to previous conclusions. The possible reason is that in addition to the size or proportion of green space, the quality of green space also has a significant impact on pollinator visitation.” Can you elaborate more on the differences in the quality of green space of your study sites that led to the unexpected negative effect of green space on pollinator visitation? Is it related to management, neighboring areas, local plant richness/composition/abundance?

Lines 312-325 – The authors acknowledge that the potential effect of night-light on diurnal pollinators, may in fact be due to urbanization (correlated with night light). However, they discuss the potential direct effects of night-light to diurnal pollinators (see refs 26,47) and this can be better explained.

Line 354-357 – The authors state that “In urban green space, the application of core flowering plants plays an important role in mitigating the negative effects of urbanization and maintaining pollination services [38]. The use of core flowering plants can help improve the quality of green space and promote the function of green space in protecting biodiversity.” Using your findings you should indicate some recommendations for the improvement and management of green space in cities. The exotic plants you identified as core should be planted or instead you should use native species from the same genera? What native plant species have you found (and also from other studies in China) that should be used for pollinator conservation in cities? Please address this important topic in the manuscript.

Line 358 - In section 6 you repeat the information on results and provide no discussion on the subject. Probably you do not need to discuss this topic as it is very well known and you add nothing different … better focus on the effects of urbanization and recommendations for green space planning and management in cities.

Line 369 – Conclusions

I disagree with your conclusions: your recommendations for green space planning and management are poor and not clear, they need a better scientific support (by using other findings on plant-pollinator interactions in China) and they are problematic since you consider using exotic plants without even considering the local plants. The authors should conclude the discussion with their recommendations for green space planning by developing this important topic. Your recommendations need to be objective, based in your data but also in other studies in China that address pollination, native flora, and gardening (you need to value these local studies). The selection of plants for gardening needs to be rigorous if you aim to preserve native pollinator diversity. The use of exotic plants may favor exotic pollinators and contribute to native pollinator decline.

Reviewer 2 Report

The authors investigated the influence of various factors of the urban environment on the diversity and abundance of flower visitors from different functional groups. These analyses were made according to the urbanisation degree and confirmed the significance of several factors. In addition, the authors created a network of relationships between flower visitors and plants.

But, some explanations of the study methods are necessary.

The authors selected 11 functional groups to show the network of relationships between plants and flower visitors. The group Apoidea is divided into the honey bee, Halictidae, Anthophora spp. as long-tongued bees. Why Bombus and other wild bees were omitted? What number of species were detected in the study, and how does this represent the apidofauna of the region existing in an undisturbed environment.

Why are the relationships between plants and functional groups described with binary networks but not between plant species and insect species? Was the network of plants and visitors created based on data collected from all zones of the cities? Maybe Shannon’s diversity index of interactions (calculated as a proportion of interactions) estimate for each city zone separately will show differences in the network of relationships?

In Fig. 4 or Appendix B, is it possible to indicate in which city zones (Core, Suburb, Exurb) plant species were found?

I suggest using the “impervious surface” (which is a well-defined concept) as a factor instead of the urbanisation level. The study aims to examine individual factors. Buildings are essential for urban development but, the primary indicators of urbanisation are population density and the number of inhabitants. 

Chapter discussion needs improvement for better interpretation of results. The authors cite the results of other authors but do not try to explain the differences and often repeat information from the Results section. For example, lines 272-276; 344-345. Also, subsection 4.6 contains repeated information from the results section, except for the last sentence (line 366), which is a very general statement.

The conclusion is too speculative. The diversity of plant species was not high, and the choice of forage plants for insects was limited. If these four species dominated the occupied area, this was instead the reason for the frequency of flower visits, not the feeding preferences of the insects.

Specific comments

Line 262-264 The sentence written and the paper cited (#40 and #41 from the reference list) are irrelevant

Line 304-305.  Muscidae are a different ecological group of insects and have various habitat requirements than bees. So, other factors can cause changes in the fly population.

Line 304. “...which was contrary to previous conclusions.” It would be necessary to cite the paper after this sentence; otherwise, it points to the conclusion in the previous sentence (in line 303).

Line 308.  Do you mean the areas occupied by cut grasses and anemophilous plants?  To make such a conclusion, it would need to be more specific about the type of plant community of the study plots. This is also not consistent with the information you mention in the methods, that plots without flowers were not taken into calculation.

Line 324. If you explain this by human activity, you negate light’s effect. How to distinguish the direct effect of light on Halictidae from the impact of the urbanisation level, with which artificial lighting strongly correlate probably?

Reviewer 3 Report

please see provided document

Round 2

Reviewer 1 Report

The authors made changes that significantly improved the manuscript. Their work is informative, easy to read, findings were better presented, analyzed and discussed, the conclusions are supported by the data and the overall context was better addressed. Also very important,, the study limitations were stressed and caution was adopted for data interpretation.

In the revised version I found that urbanization levels are missing in figure 4 (figure and legend) and just made two minor suggestions in the text. 

Now that I saw the sampling design (figure 2), I need to tell that it is not the most adequate for sampling mobile organisms (as pollinators) since in the 200m transect the largest distance between observations were ~20m and many observations were made very close (lines distanced by 2m only). To ensure the independence of the data (i.e. not counting the same individual twice or more) transects should ideally be linear or use another approach (count data on focal plants).

Congratulations for your work, Best wishes!
